# Bone Marrow Multipotent Mesenchymal Stromal Cells as Autologous Therapy for Osteonecrosis: Effects of Age and Underlying Causes

**DOI:** 10.3390/bioengineering8050069

**Published:** 2021-05-17

**Authors:** Jehan J El-Jawhari, Payal Ganguly, Elena Jones, Peter V Giannoudis

**Affiliations:** 1Department of Biosciences, School of Science and Technology, Nottingham Trent University, Nottingham NG11 8NS, UK; 2Clinical Pathology Department, Mansoura University, Mansoura 35516, Egypt; 3Leeds Institute of Rheumatic and Musculoskeletal Medicine, School of Medicine, University of Leeds, Leeds LS2 9JT, UK; P.Ganguly@leeds.ac.uk (P.G.); E.Jones@leeds.ac.uk (E.J.); pgiannoudi@aol.com (P.V.G.); 4Academic Department of Trauma and Orthopedic, School of Medicine, University of Leeds, Leeds LS2 9JT, UK

**Keywords:** multipotent mesenchymal stromal cells, osteonecrosis, autologous bone marrow, regenerative therapy

## Abstract

Bone marrow (BM) is a reliable source of multipotent mesenchymal stromal cells (MSCs), which have been successfully used for treating osteonecrosis. Considering the functional advantages of BM-MSCs as bone and cartilage reparatory cells and supporting angiogenesis, several donor-related factors are also essential to consider when autologous BM-MSCs are used for such regenerative therapies. Aging is one of several factors contributing to the donor-related variability and found to be associated with a reduction of BM-MSC numbers. However, even within the same age group, other factors affecting MSC quantity and function remain incompletely understood. For patients with osteonecrosis, several underlying factors have been linked to the decrease of the proliferation of BM-MSCs as well as the impairment of their differentiation, migration, angiogenesis-support and immunoregulatory functions. This review discusses the quality and quantity of BM-MSCs in relation to the etiological conditions of osteonecrosis such as sickle cell disease, Gaucher disease, alcohol, corticosteroids, Systemic Lupus Erythematosus, diabetes, chronic renal disease and chemotherapy. A clear understanding of the regenerative potential of BM-MSCs is essential to optimize the cellular therapy of osteonecrosis and other bone damage conditions.

## 1. Introduction

Bone marrow-multipotent mesenchymal stromal cells (BM-MSCs), the key bone regenerating cells, were first identified as attached to plastic culture surfaces and characterized by a self-renewal capacity to form a colony and the ability to differentiate into bone, cartilage and fat cells [1]. With accumulating scientific knowledge on the MSC roles in tissue regeneration, their other functions, including immunoregulation, proliferation, angiogenesis support and pro-survival abilities, became important to consider for application in regenerative therapies [2,3,4]. The best-characterized and most frequently used source of MSCs is BM, even with the very low frequency of these progenitor cells, i.e., 0.01–0.001% of nucleated BM cells [1,5].

Osteonecrosis (ON), also named avascular necrosis, is a multifactorial and painful bone disease characterized by vascular insufficiency, progressive collapse of subchondral bone and cartilage damage. This disease often develops in the femoral head, but it could also affect other joints, mostly in young or middle-aged individuals. One main mechanism of ON pathogenesis is vascular damage due to clotting, embolism, increased intraosseous pressure and direct blood vessel injury [6]. ON can also be caused by direct trauma or infections. The most common non-traumatic conditions that can trigger ON are corticosteroids, alcohol abuse, Gaucher’s disease, sickle cell disease, Systemic Lupus Erythematosus, diabetes, chronic renal diseases and chemotherapy (Figure 1) [6]. Over the last three decades, autologous MSCs have attracted considerable interest as cellular regenerative therapies for ON [7]. For enhancing therapeutic outcomes, it is also important to consider the effects of patients’ age as ON is common among young and old populations but can occur in elderly individuals according to the underlying conditions [7,8,9]. Here, we review how these predisposing causes of ON and age can affect the biological characteristics of MSCs.

## 2. MSC Therapy for Osteonecrosis

### 2.1. Pre-Clinical Studies

Extensive experimental research (in vitro and in vivo studies) has confirmed the unique and multifaceted regenerative capacities of MSCs. Because of their unique biological properties, MSCs represent a promising therapeutic tool for ON (Figure 2). Importantly, MSCs provide angiogenic support function mostly due to the production of Vascular endothelial growth factor (VEGF) and control osteoclastogenesis via the release of soluble factors such as RANKL and OPG [10,11]. These cells also have essential differentiation capacities needed to repair damaged bone and cartilage tissues in ON [12]. More recently, their high proliferation and ability of suppression of immune response associated with tissue damage via the production of immunosuppressive factors, indoleamine 2,3 dioxygenase (IDO), Transforming growth factor-beta (TGF-β), prostaglandin E2 (PGE-2) and IL-10 have been considered equally important [13,14,15]. A controlled inflammatory reaction is required for the normal bone healing process and is deemed necessary for timely fracture repair and tissue regeneration in ON [14]. 

An important factor in the therapy of ON is the use of 3D scaffolds that are used as bone fillers and additionally support the repair functions of MSCs [16]. These scaffolds also overcome the disadvantages of autografts and allografts, such as limited accessibility and side effects of immune rejection and infection [16]. The needed criteria for scaffolds used in regenerative therapies include biocompatibility, biodegradability, porosity, and mechanical support to avoid bone collapse during the repair process. Several natural and synthetic materials are used to fabricate scaffolds [16,17]. Polymers are commonly used in ON treatment include as poly lactide-co-glycolide (PLGA), poly ε-caprolactone (PCL), Cervi Cornus Colla (CCC), polyethylene glycol maleate citrate (PEGMC), polylactide (PLA), polymethyl methacrylate (PMMA), and peptide-based hydrogels, but they lack mechanical stability [16]. Other materials include natural polymers, such as hyaluronic acid and collagen reinforced with demineralized bone matrix (DBM), and inorganic components of bone, e.g., calcium phosphate (CP), β-tricalcium calcium phosphate (TCP) and hydroxyapatite (HA) [16]. Using porous HA or TCP scaffolds together with BM MSCs enhanced bone regeneration in osteonecrosis, as seen in experimental models [18]. Popular materials for synthetic scaffolds also include ceramics, bio-glass and porous titanium (Ti) [16]. 

In a composite approach, bioactive cells and growth factors are included to ‘functionalize’ the scaffolds and enhance the osteoinductivity. A combination of scaffolds and MSCs is a popular method to repair necrotic areas, as the scaffolds help direct MSCs to the necrotic areas and deliver mechanical support for these cells. Ceramic scaffolds loaded with BM-MSCs promoted the healing of bone defect in a canine model of femoral head ON [19]. Porous tantalum scaffolds were used to load MSCs following surface cover with Bio-Gide^®^ collagen membrane [20]. This hybrid therapy has helped induce bone and cartilage repair of femoral head necrosis in a rabbit model [20]. The seeding collagen scaffold with hypoxia-pretreated BM MSCs induced bone regeneration and angiogenesis in rabbit femoral head ON as hypoxia increase viability and osteogenic capacity of MSCs [21]. In addition to MSCs, more complex constructs can be potentially used as scaffolds loaded with both adipose-derived MSCs and endothelial cells, shown as a potential therapy for ON in a rat model [22]. 

Another successful approach for ON therapy is the use of scaffolds functionalized by growth factors to ensure the sustained release of these factors, which would enhance MSC regenerative functions and blood vessel formation. This combination also solves the problems of local application of growth factors, such as their short half-life in vivo or heterogenous ossification related to the applications of BMPs [23]. Encapsulation of growth factors, such as BMP-2 or VEGF [24,25] in the PLGA microsphere, showed great results in experimental models of bone necrosis. Combined platelet-rich plasma (PRP) with scaffold has been shown to have the capacity to heal defects in a rabbit model of ON [26]. Another potential tool includes platelet-derived growth factors that can be used in different formats. Allogeneic platelet lysate or PRP prepared from young, healthy donors has a great potential enhancing chondrogenic and osteogenic potential of MSCs. This combination therapy is popularly considered for traumatic bone damage and various degenerative joint diseases [27,28]. In a case study, a combined local injection of adipose tissue-MSCs with platelet-rich plasma showed satisfactory outcomes in an experimental model of femoral head ON [29]. Another approach for enhancing MSC-based therapy is the use of genetically modified MSCs. The bFGF- gene overexpressed lentivirus-transfected BM MSCs combined with cancellous bone is one example that was tested in a rabbit model of ON with bone healing success [30]. Furthermore, synthesized adenovirus-mediated BMP-2 and bFGF modified BMSCs combined with DBM were effective in the repair of a canine femoral head defect [31].

### 2.2. Clinical Studies

Initial studies on using autologous MSCs for the treatment of ON were based on cultured MSCs from bone marrow (BM) aspirates and have demonstrated excellent patient safety and efficacy profiles as proved by pain relief and radiological signs of healing [32,33]. A phase I/II study in sickle cell disease patients who had ON showed a high success rate when using autologous MSCs up to 60 months of follow-up [34]. Similarly, Gangii et al. showed that autologous BM-MSCs were effective to treat the early stage of ON for the 60-month follow up [35]. In common, the outcome of MSC-based therapy shows better outcomes in the early stages of ON and traumatic rather than non-traumatic patients [36,37].

Using uncultured autologous MSCs are particularly preferred for ease of extraction, safety and minimal need for handling, processing, and ethics [7]. Pioneering work by Hernigou et al. of using BM autologous MSCs to treat patients with femoral head ON in combination with core decompression has indicated the potential therapeutic value of these cells, particularly in a dose-dependent fashion [38]. The same group showed that the numbers of BM MSCs are reduced in ON and suggested better techniques to increase the yield of harvested progenitor cells, such as optimizing the BM aspiration method and using BM concentrates [39]. While the benefits of autologous BM MSCs in ON seem evident, more focus on assessing the numbers and, particularly, the biological fitness of therapeutic autologous MSCs is needed to develop these therapies further. 

Core decompression is a classical surgical treatment for ON but usually effective for small necrotic lesions. The combination of core decompression with autologous BM was associated with improved clinical scores and radiological signs [40]. Some clinical studies have included other supporting factors for MSCs. Injection of concentrated BM and platelet-rich plasma was used for 77 ON patients and showed clinical improvement in 86% of the tested group [41]. In another study, a combination of core decompression, autologous BM-MSCs and calcium sulphate/calcium ceramic scaffold showed satisfactory outcomes. However, that study lacked controls of single therapeutic tools [42]. Other functional scaffolds were tested, and Kuroda et al. showed that using slow-release scaffolds to deliver human FGF was an advantage to promote regeneration of bone necrosis in femoral ON patients [43].

The BM is a common source for MSCs used for ON, and the clinical delivery of BM MSCs as mononuclear cells was found to be better than using whole BM, likely because of concentrated cells [44]. Adipose stromal cells were also applied into hip joints and resulted in positive clinical outcomes [45]. In another study, a combination of umbilical cord MSCs and BM mononuclear cells improved ON patient symptoms [46]. Furthermore, the delivery of MSCs is commonly directed into the area of ON, but the systemic infusion could be another route of delivery. Co-infusion of BM cells and umbilical cord MSCs through femoral artery resulted in clinical improvement as conducted in Phase I/II clinical study on 30 femoral head patients [46]. Additionally, Mao et al. found that intra-arterial MSCs was effective in treating early-stage ON [47]. However, additional research is still required to determine whether the local delivery or the systemic infusion of MSCs is more effective in ON therapy.

In summary, the great potential of autologous MSCs alone or in combination with scaffolds and growth factors for tissue repair in ON is evident [6,48]. However, different ON-underlying factors could hamper the regenerative potential of autologous MSCs. As there are multiple causes for the development of ON, it is essential to assess the patient’s suitability for autologous MSC use carefully and consider how the underlying cause of ON might affect the biological fitness of these regenerative cells. This knowledge will help to optimize autologous MSC-based regenerative therapies for ON. Furthermore, the effect of a patient’s age should be taken into account based on the recent advances in this area of research.

## 3. Age-Related Changes in MSCs

ON is common in the young/middle age group [7,49]. While chronological aging is related to the numbers of years, the process of biological aging can be promoted by environment, diet, disease, heredity and lifestyle [50]. At the cellular level, aging is a complex process caused by an accumulation of various types of damages in cells over time [51,52]. The free radical theory refers to the damage caused by intermediate oxygen molecules resulting from cellular metabolism. In older age, oxidative stress outgrows the level of anti-oxidant enzymes resulting in increasing levels of reactive oxygen species (ROS) [53]. Another theory suggests that replication errors and extrinsic factors (such as radiation and ROS) progressively cause damage to the DNA, ultimately leading to cell senescence or death [50,54,55]. These general theories apply to MSC ageing characterized by a decline in their number [56], increased ROS leading to a shift towards adipogenic differentiation [57], a decline in telomere length [58], as well as increased DNA damage [59]. 

### 3.1. Pre-Clinical Studies

Previous in vitro studies have used the long-term passaged MSCs in vitro as an indication of ageing. These studies showed that several changes are associated with in vitro ageing of MSCs [60]. These changes include reduced survival and proliferation, increased senescence and ROS expression levels, decreased differentiation potential, and increased genetic instability [60]. However, recent studies on MSC healthy aging in vivo (as discussed below) could change our understanding of aged MSCs and their use in therapy for ON.

### 3.2. Clinical/Human Studies

The CFU-F assay has been the oldest method for enumerating BM-MSCs by counting the numbers of single cell-derived colonies based on first the MSC ability of plastic-adherence and second their ability to proliferate forming a colony. Over the years, several groups have performed CFU-F experiments to examine the age-related changes in the number of aspirated iliac crest BM-MSCs (Table 1). The variable results obtained can be due to different factors. For example, different volumes of BM aspirates have been used (Table 1), but larger-volume aspirations could lead to the dilution of BM aspirates with blood [61] and subsequently produce artificially low MSC frequencies, as MSCs are not present in the peripheral blood [62,63]. Further processing of the BM aspirate for MSC isolation as well as using different media and different scoring criteria, could also impact the CFU-F frequency [64,65,66] (Table 1).

Irrespective of these compounding factors, a general trend for a decline in BM MSC numbers has been confirmed in several recent studies. Most recent data from our laboratory (n = 67 donors, 33 females and 34 males) indicated a significant age-related decline in the number of colonies in relation to donor age in both males and females [73]. Interestingly, while the overall median CFU-F frequency was the lowest in the old donor group (61–89 years old), the decline was the steepest between young donors (19–40 years old) and the donors of the intermediate age group (41−60 years old) [73]. Similarly, another study has found a decline with age, but no groups were analyzed [74]. This was also noted in the study by Siegel et al., which implied that the decline in MSC numbers begins in the fifth decade of life [56].

Enumeration of uncultured BM-MSCs has also been attempted by flow cytometry using several surface markers [75,76,77]. CD271 (low-affinity nerve growth factor receptor) in combination with CD45 (pan-hematopoietic lineage cell marker) has been recently reported to provide the best gating strategy for BM-MSCs [77,78,79]. In two of our recent studies, BM-MSCs were quantified volumetrically using the CD45^low^CD271^+^ phenotype [72,73], and in both studies, we observed trends for age-related decline in the numbers of MSCs [72,73]. However, a steeper decline was observed in the intermediate age group (41–60 years old) as compared to the young donor group (19–40 years old), rather than in the donors of the old age group (61–89 years old) compared to the intermediate age group, in agreement with CFU-F findings [73]. Overall, these studies indicate that autologous MSCs in relatively young ON patients are likely to be higher in number compared to those from the middle-age group.

Using uncultured MSCs, our group has recently shown no significant age-related increase in adipogenic or osteogenic differentiation transcripts or ROS levels in uncultured MSCs from old donors compared to young donors [73,80]. This contrasts with the higher ROS levels found in minimally cultured MSCs from older donors [81]. Very recently, another group evaluated the number and functionalities of MSCs from young and old donors and found no age-related differences in growth kinetics, tri-lineage potential, gene expression profiles and immunosuppressive properties [82]. When the same biological characteristics were compared between early and late passage MSCs, they displayed significant differences. Although this indicted little implication of donor age on MSC immunomodulatory functions, more research would be needed to assess this function in other ON-underlying diseases. This research will be particularly important for therapeutic MSCs as defective immunosuppression has been recently reported with a link to complicated bone healing [83].

The age-related changes of the extra cellular matrix (ECM) could affect the functions of MSCs and their use for ON therapy [84]. The ECM is formed of proteoglycans, fibrous proteins (mainly collagens), minerals and water. In joints, proteoglycans absorb water and act primarily to resist load compression and create flexibility. In contrast, collagens enable cartilage to resist sheer stress. With aging, changes in these ECM components can lead to osteochondral tissue damages. As the age increases, the molecular weight of proteoglycans is decreased, and the proteolytic enzyme activity is increased [85,86], leading to reduced proteoglycan aggregates and higher levels of serum aggrecan fragments [87]. Similarly, thickness and cross-linking of collagen type II fibers are detected with aging [85]. Furthermore, an age-mediated increase in the expression and activity of collagenase activity by chondrocytes could lead to articular surface fibrillation and erosion [88]. Such disturbance in ECM homeostasis could affect the reparatory performance of intrinsic or implanted therapeutic MSCs as ECM is essential for cell survival, proliferation and communication. Collagen type II enhances chondrogenesis and osteogenesis of MSCs by facilitating osteogenic marker RUNX2 stimulation via the integrin α2β1-FAK-JNK signaling pathway [89]. Another component of ECM proteoglycans, Syndecan-1, is involved in osteo-adipogenic balance during the early induction of MSC lineage differentiation [90]. In addition to ECM, an inflammatory microenvironment such as that seen in osteoarthritis can also negatively affect MSC function by suppressing chondrogenic and osteogenic differentiation [91]. Consequently, it is important to consider the joint microenvironment to optimize MSC-based therapy for ON, particularly in aged patients. Suggestions of using young ECM to rejuvenate therapeutic MSCs also necessitates further investigations [92,93].

Summarizing the current literature in relation to age-related changes in BM-MSCs, it becomes apparent that young individuals are likely to have several-fold more MSCs in aspirated BM compared to donors over 40 years old. This can have implications on the dose of MSCs delivered into the damaged femoral head in ON patients. However, middle-aged and older donors show significant variability in relation to their MSC numbers, possibly because their skeletal biological ages are not the same as their chronological ages and may have been affected by environmental causes listed at the beginning of this section. With regards to MSC functions, such as differentiation, anti-oxidative and bone-remodeling capacities, age-related declines reported using cultured MSCs may represent an artefact of culture-expanded in vitro, and different types of assays need to be developed for the assessment of these functions in uncultured MSCs. In relation to ON treatment with autologous MSCs, these findings indicate that MSC dose-determination based on rapid laboratory tests [72], rather than ‘predicted’ from patients age, is needed for standardization of these therapies.

## 4. The Effect of Glucocorticoids on MSCs

Physiological Glucocorticoids (GC) hormones are produced by the adrenal cortex, bind to their cell receptors and, consequently, transport to DNA, modifying cell biology. Although GC therapies are commonly used for anti-inflammatory effect, their significant immunomodulation could trigger undesirable effects that impair bone biology, causing osteoporosis [94]. Additionally, GC-based therapy is a major cause for developing ON, with a 5–25% prevalence among non-traumatic ON cases [94,95]. The risk of developing ON is known to increase with long-term, high-dose and intra-articular administration of GC [96,97]. Different mechanisms of GC-induced ON have been described, particularly the decrease of the production of osteoblasts and osteoclasts, the induction of the apoptosis of both cells and the damage to the blood supply of bones [96,97]. Additionally, it has been shown that patients who have these medications display reduced BMP2 and osteocalcin gene expression in BM stromal cells and are very likely to develop ON [98].

### 4.1. Pre-Clinical Studies

The effect of GC on MSC biology was investigated in the cells of several experimental models as well as human cells ex vivo. BM-MSCs of age-matched rats were treated in vitro with various concentrations of 17 β-estradiol and dexamethasone. Interestingly, the data showed that these steroids changed MSC proliferation and differentiation in a dose-dependent manner. A low dose of these steroids helped to enhance MSC proliferation and osteogenic differentiation as noted by an upregulation of alkaline phosphatase (ALP), osteocalcin (OCN) and calcium levels as well as TGF-β1 and BMP-7 transcripts. However, a high dose of these steroids inhibited MSC differentiation and proliferation [99]. Therefore, low doses of dexamethasone are commonly used in specialized culture media to enhance the in vitro osteogenic, adipogenic and chondrogenic differentiation of MSCs [100]. Interestingly, effective doses to enhance MSCs osteogenic potentials and growth factor expression were different between males and females and likely related to the different expression levels of steroid receptors between both genders [99]. The gender-related differences have been noted for brain and endothelial progenitor cells [101,102]. However, further investigations for such receptor differences, and consequently, therapeutic consideration would be needed for human BM-MSCs. Nuzzi et al. have tested the effect of different GC drugs, including triamcinolone acetonide, alcohol-free triamcinolone acetonide, micronized intravitreal triamcinolone and dexamethasone on human BM-MSCs. Similar to animal studies, GC exerted a negative effect on the viability and growth rate of healthy BM-MSCs, as well as changes in the cell morphology [103].

GC was also found to upregulate the Dickkopf1 (DKK-1) levels that inhibit the osteogenic Wnt/β-catenin signaling pathway in BM-MSCs, leading to bone resorption [104]. Similar to BM-MSCs, adipose tissue-derived MSCs exposed to GC demonstrated a lower ALP activity, with high serum DKK-1 levels detected in steroid-induced ON patients [104]. Interestingly, a miRNA microarray analysis on ON-MSCs or GC-treated MSCs showed that miRNA-708 was upregulated and associated with the suppression of SMAD3 and RUNX2 expression, and consequently, the osteogenic potential of MSCs, but with an enhancement of their adipogenesis [105] which might contribute to fatty bone marrow, a typical character of ON. GC can also inhibit the immunoregulatory effect of MSCs. An in vitro work showed that dexamethasone could reverse MSC-mediated inhibition of T-cell proliferation in a dose-dependent manner [106]. This dexamethasone effect is mediated by interfering with STAT1 phosphorylation and subsequently cause a reduction of an anti-inflammatory mediator, inducible nitric oxide synthase (iNOS) [106]. Furthermore, a mouse model of liver fibrosis showed effective anti-inflammation signs such as decreased T-lymphocyte infiltration when treated by MSCs alone. However, simultaneous administration of dexamethasone abolished these therapeutic effects of MSCs [106]. Control of inflammation at the early phase of bone healing, in which MSCs are involved, is essential to progress into an effective healing phase [107,108,109,110,111,112]. Considering that MSCs and other immunosuppressants could counteract each other, particularly for immunoregulation, simultaneous application of MSCs with GCs might have an adverse effect on bone repair.

### 4.2. Clinical/Human Studies

Human BM samples from individuals with GC-induced ON had lower CFU-F numbers compared to two groups of those with ON related to sickle cell disease and healthy controls [113], further confirming the suppressive role of GC on MSC proliferation.

In patients receiving systemic GC, a significant drop of serum sclerostin, OPG and OCN levels were observed. However, an increase of type 1 collagen cross-linked C-telopeptide, which were positively correlated with DKK-1 levels, were noted [114]. Together, these data suggest a negative effect of GC on bone formation potential. Effects of excess doses of GC are equally seen in patients with Cushing syndrome with increased fat deposits and osteoporosis [115].

Overall, a systematic effect of GC treatments on BM-MSCs is highly probable, potentially causing a reduction of their numbers in vivo. Nevertheless, more studies are needed to confirm this GC effect, particularly considering additive aging-related reduction of MSC numbers. In addition, impairment of repair function may be expected; therefore, boosting MSC osteogenic pathways are highly desirable in these cases. For example, targeting DKK-1 in MSCs before being used for regenerative purposes could be the solution. It is also of value to include other tools of therapeutic regeneration, such as tissue growth factors. For example, using bFGF releasing scaffolds in the dexamethasone-induced ON rat model allowed increased cartilage regeneration without cytotoxic effects on MSCs [116].

## 5. Alcohol Effects on MSCs

Chronic and heavy alcohol intake is a high-risk factor for secondary osteoporosis, bone mineral density loss, and bone remodeling impairment. Consolidate evidence showed that bone turnover and healing is negatively affected by alcohol [117,118]. For ON, alcohol is one of the common non-traumatic underlying causes. Chronic alcohol consumption is usually associated with an increase in serum triglyceride and cholesterol levels, and these could lead to the death of bone-forming cells and disturbance of bone remodeling and, eventually, osteonecrosis [119].

### 5.1. Pre-Clinical Studies

Studies on in vitro expanded human BM-MSCs indicated the damaging effects of alcohol on various MSC biological functions, particularly proliferative capacity that can translate into low in vivo counts. Alcohol induces senescence in BM-MSCs, which gradually lose self-renewal potential and are known to exhibit impaired multi-lineage differentiation potentials that will reduce the rate of bone formation and raise the osteoporosis risks [120,121]. Chen et al. tested the culture-expanded BM-MSCs exposed to a high ethanol concentration and showed a reduction of the proliferation and increase of the senescence of these cells [122]. The reduction of MSC count in cultures was measured by microscopy together with G0/G1 cell cycle arrest and upregulation of cell cycle regulators, p16INK4α and p21 levels. These alcohol-treated MSCs also showed premature senescence in a dose-dependent manner with an increase in β-galactosidase (β-gal) expression detected by fluorescence microscopy and changes in senescence-related genes; a reduction of silent information regulator Type (*SIRT1*) gene, which also controls the osteogenic commitment of MSCs and an increase of proliferation controlling gene, *P38* [122]. Additionally, chronic exposure to alcohol in adult rat for 12 weeks demonstrated that BM-MSCs from femur and tibia and adipose-derived MSCs had low proliferative capacity, as shown by the CFU-F assays [123]. Consistently, cord blood progenitor cells displayed an increased β-gal activity and shortening of telomere linked to the decrease in SIRT1 when treated by alcohol [124]. In contrast, the overexpression of SIRT1 in these cells restored non-senescent phenotypes [124].

In terms of differentiation, chronic alcohol consumption decreases the bone-forming capacity and conversely increases the adipogenic differentiation of BM-MSCs [125]. Additionally, BM-MSCs exposed to ethanol in vitro have shown a significant reduction of collagen type I expression and ALP activity [126]. As shown in a mouse model, these effects are related to the activation of mammalian target of rapamycin (mTOR) signaling cascade, causing downregulation of RUNX2 and increase of peroxisome proliferator-activated receptor-gamma (PPAR-γ) via the activation of p70 ribosomal protein S6 kinase. Conversely, blockage of the mTOR pathway by rapamycin treatment improves alcohol-induced MSC osteogenic differentiation and osteopenia [125]. Deregulation of Wnt signaling due to alcohol exposure could be another mechanism of impairment of MSC differentiation, and an incomplete healing process persists up to two weeks post-fracture [127]. Chronic alcohol exposure had an effect on fracture healing with decreased mineralization, as reported in different experimental studies [128,129,130]. Testing the mineralization using alizarin red and perchloric acid staining as well as measuring osteoblast-specific genes, RUNX2, BGLAP and COL1A1, revealed that ethanol-treated MSCs had a significantly lower osteogenic capacity due to downregulation of the SIRT1 gene [131]. Other experimental studies have reported that episodic or acute alcohol exposure in rodents negatively affects cartilaginous callus development mainly due to inhibition of canonical Wnt/β-catenin signaling [127,132,133,134,135,136,137,138]. Another mechanism of alcohol-related impairment of cartilage and bone healing is the downregulation of TGF-β1 protein expression via interference with the transcription factor myeloid zinc finger 1, as shown in vitro for human BM-MSCs [139]. Several research groups indicated that alcohol exposure could also decrease callus biomechanical strength [132,134,135]. It has also been demonstrated that mice injected with ethanol in a similar pattern of heavy episodic drinking had decreased callus size and biomechanical stiffness [140].

In an experimental study, alcoholic rats showed decreased serum levels of certain inflammatory markers, interleukin (IL)-6, IL-2, IL-10, and C-reactive protein after induction of femoral mid-diaphyseal closed fracture as well as reduction of white blood cell numbers compared to saline-injected rats [141]. Additionally, alcohol could modulate the local fracture microenvironment enhancing proinflammation, and the fracture healing in alcohol-exposed animals was found to be enhanced using IL-1 and TNF antagonists [130]. Therefore, although no studies have explored the direct effect of alcohol on the immunomodulatory functions of MSCs, it is possible that these MSC functions are, consequently, affected by alcohol-related proinflammatory changes. Regarding MSC migration, alcohol might reduce osteopontin (OPN) protein expression as well as integrin β1 receptor expression levels that are largely involved in MSC homing following bone trauma [140,142].

### 5.2. Clinical/Human Studies

One recent study has compared the numbers of MSCs in BM concentrates from three ON patient groups: alcohol, GC and trauma. Similar numbers of MSCS measured by CFU-F assay were reported for the three groups [143]. Although the study indicated not much role for alcohol and GC compared to trauma, an ideal comparison should include healthy controls. Additionally, using another method to count MSCs (e.g., CD271+ cell counts via flow-cytometry) and normalization of some differences in fold-enrichment of MSCs in BM concentrates would be needed to confirm the effect of ON causes on MSC quantity/proliferation.

In summary, chronic alcoholism has a detrimental effect on several MSC regenerative functions. Therefore, growing research aims to improve these functions. Anti-oxidant therapies, e.g., N-acetylcysteine and Vitamin D, have been proposed as therapeutic tools to prevent or minimize the negative effect of alcohol intake on bone healing, as reported in animal studies [144,145]. While the effects of chronic alcohol intoxication on bone remodeling are not permanent and can be improved after two years of discontinuing alcohol consumption stopping [146], a further understanding of the molecular mechanisms involved in the development of alcohol-induced effects may help to identify new therapeutic targets to optimize MSC-based therapies.

## 6. MSCs in Sickle Cell Disease

Sickle cell disease (SCD) is a red cell disorder associated with various vascular pathologies, capillary damages and a high prevalence of bone involvement [147]. Adesina et al. have reported that a 22% percentage of SCD patients can develop ON, particularly in patients with severe forms of the disease, in elderly patients and those with comorbidity of acute chest syndrome [148]. The pathogenesis of SCD-related ON is mostly related to the clotting or the occlusion of micro-vessels of bones [149,150].

### 6.1. Pre-Clinical Studies

The experimental mice model for SCD is usually used to understand the pathogenesis and test the gene therapy for this disease [151]. No animal studies testing SCD effect on MSCs.

### 6.2. Clinical/Human Studies

In one study, the proliferation of MSCs from SCD patients was similar to that of healthy MSCs as tested by the in vitro doubling time and the surface marker phenotype. Isolated MSCs from BM aspirate concentrates of these patients maintained their proliferative potential and chondrogenic, adipogenic and osteogenic differentiation potential [12]. Furthermore, SCD culture-expanded MSCs were found to produce osteogenesis cytokines and growth factors, IL-8, TGF-β1, SDF-1 and VEGF [34]. Hypoxia-preconditioning of SCD BM-MSCs was found to increase the production levels of trophic factors (VEGF, IL8, MCP-1 and ANG) mediating angiogenesis and tissue repair [152].

In addition to repair functions, SCD MSCs have also been shown to preserve immunoregulatory capacity, with IDO found to be the major immunoregulatory mediator [151]. Interestingly, CFU-F counts and CD271 + CD45-/low cells in BM concentrate were significantly higher in SCD than in SCD-unrelated ON patients [153]. However, the authors reported an age-related reduction of these numbers. Additionally, similar differentiation potential and secretion of cytokines were noted between the two groups [153]. A recent study by Daltro et al. showed that pediatric SCD has, on average, 27 CFU/10^6^ nucleated BM cells, with a satisfactory outcome when used for ON therapy [154]. In view of these data, it could be proposed that the younger the SCD patients, the better outcome of using autologous BM MSCs. However, further studies comparing matched-age groups would be needed to confirm the effect of SCD on MSC biology.

Considering their preserved functionality and proliferative potential, autologous MSCs were used in phase I/II, non-controlled studies of 89 SCD patients with ON via local implantation of BM aspirate concentrate [34,155]. The clinical data showed that SCD patients who received this cellular treatment had a significant pain relief measured by the Harris Hip score and without deterioration of early stages of ON [34,155]. Together, these data proved the safety and effectiveness of the BM-MSC-based autologous therapy, particularly for the early stages of ON in SCD patients.

## 7. MSCs in Gaucher Disease

Gaucher disease (GD) is commonly associated with bone abnormalities such as osteoporosis, ON or pathological bone fractures [156]. The mechanisms of ON-related bone abnormalities include the accumulation of abnormal cells which infiltrate the BM compartment, causing bone damage [157].

### 7.1. Pre-Clinical Studies

An experimental study using a mouse model of GD demonstrated that the skeletal manifestations of this disease are connected to an impairment of MSC proliferation. Additionally, a decrease in the osteogenic potential of GD MSCs in this model was shown by low ALP and RUNX2 gene expression levels [158].

### 7.2. Clinical/Human Studies

MSCs from one adult patient with GD was tested for the phenotype and functions of MSCs. These MSCs displayed standard MSC surface markers as well as normal osteogenic and adipocytic differentiation and growth. However, these MSCs showed an altered inflammatory secretome with a marked increase in COX-2, PGE2, interleukin-8 and CCL2 production relative to the normal controls [159]. These preliminary data indicated an abnormal immunoregulatory role of GD MSCs that could reflect on bone healing. Furthermore, a larger study including 10 patients reported that GD MSCs showed a defective proliferative potential, cell cycle abnormalities, and importantly, impaired osteogenic differentiation as measured using ALP activity and OPN levels. Additionally, GD MSCs have significantly upregulated levels of osteoclast inhibitor and bone lytic mediators, PGE2 and DKK-1, respectively [160].

Together, these data suggest defective proliferation and differentiation. More pre-clinical and clinical studies on the potential use of autologous MSCs in the treatment of GD-related ON are highly needed.

## 8. MSCs in Systemic Lupus Erythematosus

A high prevalence of ON in Systemic Lupus Erythematosus (SLE) has been reported, with nearly 37% of lupus arthritis patients eventually developing ON [8]. The main mechanisms that lead to ON are believed to be the therapeutic use of GC [161], immunosuppressive drugs [161,162], altered lipid metabolism and thrombophilia due to anti-phospholipid antibodies [8].

### 8.1. Pre-Clinical Studies

BM-MSCs from a lupus mouse model have shown proinflammatory rather than inhibitory phenotype when stimulated by IL-1 and TNF-α. In these MSCs, there was an increase in the expression levels of CCL19, VCAM1, ICAM1, TNF-α and IL-1β, together with induction of CD4T cells proliferation [163]. Furthermore, BM-MSCs from a lupus mouse model showed impairment in inhibiting B cell proliferation and differentiation, mediated by the downregulated CCL2 levels [164].

Several mechanisms have been proposed for SLE-related senescence and apoptosis of MSCs. The abnormalities of SLE BM-MSCs have been linked to abnormal JAK-STAT, Wnt/β-catenin, P53/P21 and PTEN/Akt signaling pathways in both human and mouse studies [165,166,167].

### 8.2. Clinical/Human Studies

Cultured BM-MSCs from SLE patients displayed early senescence by increased β-gal activity, cell cycle arrest, disordered F-actin distribution and/or downregulated telomerase activity [165,166,168].

MSCs from SLE also express standard surface markers CD44 and CD105 and lack CD14, CD34, CD45 and HLA-DR. However, these MSCs had downregulation of IL-6, macrophage colony-stimulating factor (M-CSF) and IL-7 transcripts [169], indicating an impairment of immunoregulatory functions. High serum levels of proinflammatory mediators, IL-6, IL-10 and TNF-α, are evident during SLE disease activity [170]. These cytokine alterations could be linked to decreased of MSC proliferation or osteogenic differentiation in SLE patients [171].

Shi et al. found that MSCs from SLE patients demonstrated a defective migration capacity using the trans-well assay. Additionally, fluorescence microscopy images showed an abnormal over-polymerization of the F-actin cytoskeleton, accompanied by a high level of intracellular ROS. When these MSCs were treated with anti-oxidant N-acetylcysteine, F-actin restored conformation and enhancement of the migration was noted [172]. Another mechanism included MiR-663 that could downregulate TGF-β1 and consequently contribute to inhibiting the proliferation, migration and immunoregulation functions of SLE BM-MSCs [173]. Considering their limited immunosuppressive potential, autologous BM-MSCs used for therapy of SLE were not effective in controlling the disease activity [141].

In summary, BM-MSCs in SLE patients have low proliferation, and this could affect their quantities in vivo. The use of these BM-MSCs as autologous therapy for ON should also consider the other biological defects of these cells, including reduced differentiation and migration abilities. In a recent study, culture-expanded autologous MSCs were applied to treat ON patient with comorbidity of SLE [174]. The clinical data showed no adverse effects, preserved hip function and reduced pain in these patients [174]. However, more clinical studies testing the clinical safety and effectiveness of native SLE BM-MSCs are needed.

## 9. MSCs in Diabetes

Although diabetes has not been linked to ON as an underlying factor, the simultaneous existence of both conditions is common, particularly in elderly patients or diabetic individuals who are treated with long-term GC [9]. Furthermore, diabetes has been linked to the ON of the jaw in experimental models and patients. This link was mainly explained by increased inflammation, altered immune responses and vascular damages associated with diabetes and its medications [175].

### 9.1. Pre-Clinical Studies

Several functional properties were tested for diabetic BM-MSCs and found to be defective. High serum concentrations of TNF-α in diabetic mice was directly linked to low expression of Indian hedgehog (Ihh) in MSCs and, consequently, suppression of MSC expansion and bone regenerative potential [176]. Interestingly, local delivery of purified Ihh to the fracture site via a slow-release hydrogel-rescued proliferation of MSCs [176].

Kim et al. showed that the gene expression of angiogenic factors, VEGFs and angiotensin were reduced in diabetic MSCs, which, when cocultured with endothelial cells, showed significantly lower endothelial tube formation [177]. The growth factors linked to bone healing were reduced when tested in diabetic rodents’ models. A low VEGF expression in the plasma and callus tissue, with poor micro-vessels and high chondrocyte apoptosis in the fracture callus, was seen in diabetic mice [178]. Another group reported a significant association between reduced levels of serum VEGF and delayed fracture healing in diabetic rats [179]. Other growth factors involved in bone healing FGF-2 and IGF-1 were significantly decreased in serum and callus of tibial fracture in diabetic rats. These rats also displayed a lower density and area of new bone [180]. However, expression levels or response of MSCs to altered levels of these growth factors in diabetic patients remain to be established.

The osteogenic differentiation of MSCs in a diabetic rat was reported to be impaired compared to control MSCs, as measured by calcium deposition [177]. Furthermore, impaired bone healing, together with lower OCN levels, was found in a critical-sized bone defect model in type 2 diabetes rats as compared to controls [181]. The delayed fracture healing in type 2 diabetes is also associated with a differentiation switch from osteoblasts to adipocytes, as seen in an experimental diabetes model [182]. The diminished osteogenic capability of MSCs was further proved by the low gene expression of osteogenic transcription factors during bone healing in experimental models [183,184]. Human bone MSCs exposed to high glucose concentration showed induction of PPAR-γ and ADIPOQ transcript expression levels during adipogenic differentiation in vitro. This adipogenesis induction was associated with a significant rise of intracellular and extracellular ROS concentrations as well as stress-related NOX4 gene under these diabetic-like conditions [185].

With regards to immunoregulation, a recent study using a diabetic mice model has suggested a link of diabetes to the altered MSC-mediated immunoregulation. Bone MSCs showed an aberrant nuclear factor-kB (NF-kB) activation, reducing TGF-β1 expression in association with defective macrophage (M2) polarization and substantially enhanced inflammation in diabetic [186]. Furthermore, inhibition of NF-kB in these MSCs or exogenous TGF-β1 treatment reversed such detrimental proinflammatory effects on fracture healing when persistent [186]. TNF-α is one of the proinflammatory markers that can be markedly increased in diabetic, hyperglycemic and insulin-resistant mice [187]. The diabetes-mediated proinflammation could trigger the chondrocyte apoptosis and, consequently, reduce cartilaginous callus [188]. Additionally, TNF-α via FOXO1 transcription factor caused an impairment of endothelial cell proliferation and downregulation of VEGF gene expression in mice with long fractured bones, impairing their healing [189].

### 9.2. Clinical/Human Studies

Compared to pre-clinical studies, less research was conducted on the characterization of human MSCs from diabetic patients. In a recent study directly comparing MSCs from type 2 diabetes to age-matched MSCs from a healthy donor, it was concluded that diabetes seems to affect the numbers but no functional capacities of MSCs [190]. CFU-F assays were used to enumerate BM MSCs, and other functional assays included proliferation, tri-lineage differentiation and angiogenesis support [190].

Additionally, MSCs from adipose tissue samples of type 2 diabetic patients induced lower suppression of PBMCs proliferation, declined expression of anti-inflammatory TGF-β1 and increased expression of proinflammatory mediators, TNF-α and IL-6 [191]. These data suggested an impairment of diabetic MSC immunomodulatory effect that can affect early phases of bone healing [83].

Collectively, diabetic-MSCs showed impairment of proliferation, angiogenesis-support, osteogenesis, enhancing adipogenesis and immunoregulatory potential. These defects should be considered to improve the outcomes for ON lesions in diabetic patients. The biological stimulation of MSCs in diabetic patients has been proposed as a potential therapeutic strategy to improve the outcomes of using these MSCs for bone healing [176]. The exposure to recombinant BMP-2/7 only or with IGF-1 prevented these changes in high glucose cultures and significantly enhanced gene expressions of type 1 collagen and OCN while suppressing that of the adipogenic marker PPAR-γ [192]. These data suggested the potential value of combining with growth factors to optimize therapeutic MSC outcomes for diabetes patients [192]. More research was conducted aiming to overcome the impaired vasculature problem during fracture healing in diabetic patients. Using VEGF and FGF helped to rescue low expression levels of OCN and RUNX-2 and to improve angiogenesis and healing of bone in diabetic mice [193]. Using microRNA to inhibit the pro-oxidant protein p66, which promotes mitochondrial ROS generation, can consequently improve the angiogenesis-supportive potential of the MSCs [194]. Another work by Bae et al. confirmed that oxidative stress caused angiogenic dysfunction in diabetic BM-MSCs. The inhibition of functionality of NADPH oxidase successfully increased angiogenic activity in these experimental BM-MSCs [195].

## 10. Chronic Kidney Diseases Effects on MSCs

Chronic kidney diseases have been linked to ON, particularly of the jaw, with aging and low renal function [196] in addition to children with chronic renal failure [197]. Different mechanisms have been proposed linking chronic kidney diseases to ON, including using bisphosphonate and immunosuppressants after renal transplantation [198].

### 10.1. Pre-Clinical Studies

BM-MSCs derived from chronic kidney disease rats demonstrated premature senescence as indicated by the upregulation of β-gal activity, spontaneous adipogenesis, reduced growth rate and accumulation of actin [199]. Additionally, the MSC immunoregulatory effect was tested in vivo in a mouse model of nephritis, showing no immunosuppressive effect, unlike healthy MSCs [199]. Further studies to assess MSC functional capacities are needed before considering autologous MSC therapy in renal disease-related ON.

### 10.2. Clinical/Human Studies

Even though several clinical studies have assessed the safety and outcomes of using autologous MSCs for the treatment of mixed cause ON [33,34,155,200,201,202,203], a special focus on individualized response to therapy when chronic kidney disease is the underlying cause is essentially needed.

## 11. Chemotherapy Effects on MSCs

Chemotherapy is one of the reasons that can increase the risk of ON, most commonly in male patients receiving chemotherapy for testicular cancer and those treated by chemotherapy for breast, ovarian, small-cell lung cancers and osteosarcoma [204]. The mechanisms include the death of osteocytes, clotting and damage of blood vessels.

### 11.1. Pre-Clinical Studies

The effect of anti-cancer chemotherapy on MSC biology has been studied in different reports. An in vitro study tested the functions of BM-MSCs that were treated with chemotherapeutic agents used to treat leukemia such as cytarabine, daunorubicin and vincristine. These MSCs showed reduced viability, osteogenic, chondrogenic and adipogenic differentiation. Interestingly, these MSCs also showed decreased expression of cell surface receptors CD90 and CD73 but increased expression of inflammatory cytokines, IL-6 and TNF-α [205]. Another in vitro study tested the effect of chemotherapeutic medications usually used for bone marrow transplantation. The data showed that human MSCs were resistant to apoptosis induced by busulphan, cyclophosphamide and methotrexate). However, MSCs were relatively sensitive to paclitaxel, vincristine, etoposide and cytarabine [206]. Nicolay et al. reported that human BM-MSCs are sensitive to bleomycin-mediated apoptosis and reduced adipogenic differentiation relative to adult fibroblasts. In contrast, the morphology, surface marker expression and migration potential showed no alterations [207]. These in vitro studies indicated variable effects of chemotherapeutic medications, but mainly on cell survival and proliferation.

Other studies investigated the biology of in vivo MSCs in host treated with chemotherapy. In an experimental study, mice were treated with busulfan, cyclophosphamide, cytarabine, methotrexate and bortezomib, as used clinically. The data showed that CFU-F numbers were reduced significantly and were not recovered after 6 weeks. However, differentiation capacities of MSCs were not affected by these chemotherapeutic medications [208]. Furthermore, it has been shown that Wistar rats that received Doxorubicin had BM MSCs with similar surface marker and viability to controls. However, these MSCs demonstrated lower alkaline phosphatase production and proliferation potential [209].

### 11.2. Clinical/Human Studies

With regards to human studies, the numbers of BM-MSCs from patients who received chemotherapy for BM transplantation were suppressed as indicated by significant lower CFU-F numbers than controls. The reduction of MSC quantity was verified to be host, not donor-related, and these patients also had a reduced bone mineral density [210]. Similarly, BM-MSCs from patients receiving high doses of chemotherapy for hematological malignancies displayed a significant decrease in proliferative capacity. However, no changes were detected in the expression levels of surface markers, CD105, CD44 and CD29. Furthermore, immunofluorescent staining of alkaline phosphatase, oil red, lipoprotein lipase, Alcian blue and aggrecan indicated preserved tri-lineage differentiation [211]. In a study where lymphoma patients received a high-dose chemotherapy conditioning regimen followed by autologous stem cell transplantation, BM MSCs showed reduced CFU-F counts but preserved morphology, immunophenotype and differentiation capacity with no differences in gene expression profile [212].

Altogether, these data suggest potential disadvantages of using autologous MSCs in chemotherapy-treated patients for further use for ON, mainly due to reduced proliferative potential.

## 12. Conclusions: Challenges and Perspectives

Treatment of ON is generally challenging due to late symptoms, complicated pathogenesis, different severity stages and multiple causes. Using MSCs in the therapy of ON is promising, with agreement on their benefits for the early-stage ON and the value of having adequate numbers, but there are still issues to be resolved [213]. The MSC-based therapy for ON lacks the standardization of isolation, expansion (if cultured cells are used) as well as transplantation/delivery methods. Another challenge is tracking and understanding the in vivo fate of therapeutic MSCs and assessing their functional competence. Both pre-clinical and clinical studies are needed to fill these gaps of knowledge and therapeutic methods, respectively.

While the numbers of BM-MSCs decline with advancing age, their potential and functionalities undergo subtle changes in vivo that might be influenced by environmental factors such as diet, exercise and other lifestyle factors. Furthermore, individuals with comorbidities, for example, diabetes, SLE and GD or those who chronically use GC or alcohol, potentially have significant defective functions of MSCs. Different functional abnormalities of MSCs were detected in secondary ON due to the etiological factors, with some reports showing no differences between the various causes. However, it is important to stress that future studies comparing the ON causes should consider the age and probably gender matching as well as other technical details, e.g., fold-enrichment in BM concentrates if these samples are analyzed for MSC quantities.

Considering the defective quality or quantity in aged or ON-related conditions, using MSCs for ON therapy is likely to require larger or concentrated volumes of autologous BM aspirate or biological stimulation prior to the application of autologous MSCs to achieve successful ON treatment. In this context, new centrifugation techniques such as vertical centrifugation could show enhanced potential to harvest more MSCs from these individuals [214]. Combined therapies for ON are vital to overcoming any defect in MSC functions in these patients in order to enhance new blood vessel, bone and cartilage formation. Growth factors and gene-editing techniques have been proposed to correct abnormalities of these autologous MSCs before implanting them into osteonecrosis areas of the joints. Bone morphogenetic proteins (BMPs) can facilitate the bone formation, and they have been shown to improve the repair functions of BM-MSCs seeded in the femoral head of rabbits [215]. Additionally, loading β-tricalcium phosphate scaffold with BMP-2-gene-transduced BM-MSCs was successful as a therapy for early-stage ON in goats [216]. In terms of gene-editing, transplantation of hepatocyte growth factor transgenic autologous BM-MSCs has been shown to promote angiogenesis and bone regeneration in a rabbit model of hormone-induced ON [217]. Furthermore, three-dimensional (3D) bioprinting technology is an emerging tool that holds great potential for regenerative therapies [218]. Using stem cells, 3D bioprinting can provide scaffolds that could deliver high attachment and growth support for MSCs in a specific size and shape that fit the defective tissues. Further attention is needed in the scaffold design to promote vascularization. Nanotechnology has been tested for gene delivery, scaffold fabrication and monitoring stem cell proliferation [219]. Using nanoparticles will also enhance the monitoring of MSC fate and promote their differentiation, thus developing regenerative therapies. However, it is still essential to assess their safety and the best choice of materials. Future studies to test the application of these stem cell-related technologies to treat ON would be of great value.

Autologous BM-MSCs can be delivered for regenerative therapies in their native form within BM aspirate concentrates or after culture expansion, aiming to achieve reparative effects with minor complications [33,200,201,202,203]. While native autologous MSCs used in BM aspirate concentrates were found to be effective and can be readily used [220], culture-expanded autologous MSCs have been proven safe and effective for treating ON in recent clinical evaluations [48]. Different tissue sources of MSCs may have better potential than BM-MSCs in conditions underlying ON, but further research is needed to confirm this idea. An example is using reamer aspirator irrigator (RIA) waste MSCs, which are abundant and can be functionally competent for bone regeneration [221]. Using allogeneic MSCs could be a solution, as suggested by Chen et al., who used umbilical cord MSCs with no side effects for ON [222], but further comparison versus autologous cells to affirm the advantages of allogeneic cells is needed. An alternative for MSCs to enhance bone regeneration is induced pluripotent cells (iPSCs). Recent reports indicated the capacity of iPSCs to differentiate into osteoblasts or osteoclasts, indicating the potential value of these cells in enhancing bone regeneration and remodeling, particularly using animal models [223]. Still, more in vivo work is required to assess the risk of tumorgenicity and genomic instability of using these cells. Additionally, more research is needed to provide safe and effective approaches for promoting the therapeutic potential of iPSCs. Finally, future research should focus on developing better animal models simulating human ON pathogenesis to enable the application of combined therapies involving MSCs, scaffolds and growth factors. A better understanding of compounding factors that meet the specific clinical status of ON patients is clearly necessary for the development of personalized therapy for these patients.

## Figures and Tables

**Figure 1 bioengineering-08-00069-f001:**
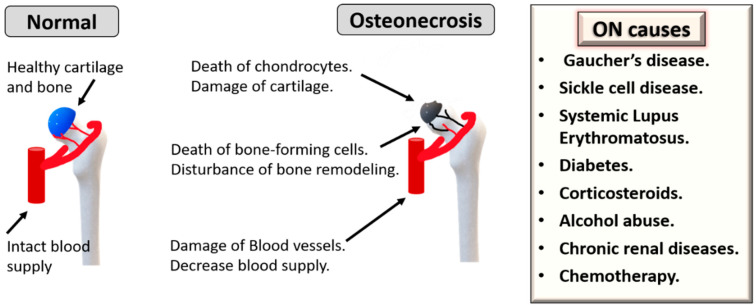
Underlying causes and pathogenesis of osteonecrosis (ON). Chronic diseases can cause ON, such as sickle cell disease, Gaucher’s disease, Systemic Lupus Erythematosus, diabetes, chronic renal disease and chemotherapy. Additionally, corticosteroids and alcohol abuse are other triggers.

**Figure 2 bioengineering-08-00069-f002:**
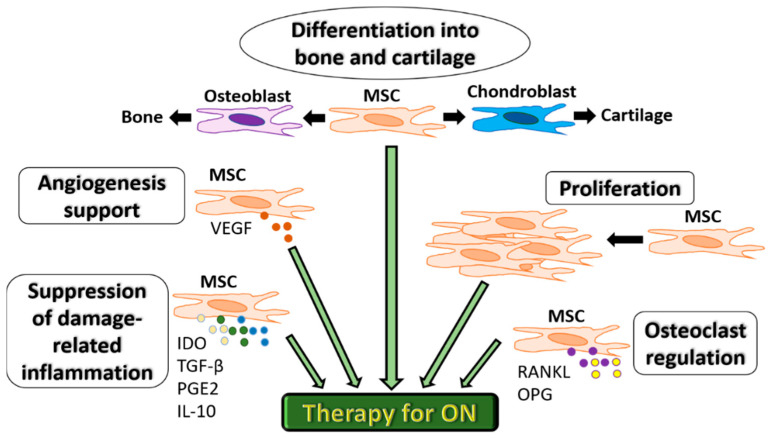
Functional characteristics of MSCs for therapeutic use in ON. MSCs can differentiate into bone or cartilage, support angiogenesis via VEGF secretion, proliferate to compensate for bone cell death, regulate osteoclastogenesis via RANKL and OPG and suppress inflammation associated with tissue damage via the production of immunosuppressive factors IDO, TGF-β, PGE-2 and IL-10.MSC: mesenchymal stromal cells. IDO: indoleamine 2,3 dioxygenase.

**Table 1 bioengineering-08-00069-t001:** List of previous studies investigating age-related changes in the numbers of BM MSCs counted by the CFU-F assay.

Age (Years)	Volume (mL)	Isolation	Media	Colony Definition	CFU-F	Ref
Y: 22–44, O: 66–74	10 mL	DC	α-MEM + 10%FCS	>16 cells	No change	[67]
13–79, no groups	4 × 2 mL pooled	DC	α-MEM + 10%FCS + ASC + Dex	>8 cells	No change	[68]
Y: 0–18, O: 59–75	NR	DC	DMEM + 10%FCS	NR	Decline	[58]
Y: 19–40, O: > 40	NR	DC	DMEM + 10%FCS	>50 cells	Decline	[69]
Y: 6–16, O: 29–76	NR	PA	DMEM + 20%FCS	NR	NS decline	[70]
1–52, no groups	NR	DC	DMEM + 20%FCS	>50 cells	No change	[71]
22–80, no groups	8 mL	PA	StemMacs medium	>50 cells	Decline in women	[72]
Y: 20–40, I: 41–60, O: >60	10 mL	PA	StemMacs medium	>50 cells	Decline	[73]
14–59, no groups	30 mL, 3 × 10 mL	PA	DMEM/Ham’s F12 + 10% FCS + bFGF + heparin	>50 cells	Decline	[74]
Y: <45, I: 45–65, O: >65	NR	DC	α-MEM + 10% human serum	>50 cells	Decline	[56]

BM: Bone marrow, I: intermediate, O: Old, Y: Young, PA: Plastic adhesion, DC: Density centrifugation α-MEM: Alpha–Minimum essential medium, DMEM: Dulbecco’s minimum essential medium, FCS: Fetal calf serum, ASC: sodium ascorbate, Dex: dexamethasone, bFGF: basic fibroblast growth factor, NR: Not reported, NS: Non-significant.

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
