# Peer review of "Bone Marrow Multipotent Mesenchymal Stromal Cells as Autologous Therapy for Osteonecrosis: Effects of Age and Underlying Causes"

_bioengineering, 2021, doi:10.3390/bioengineering8050069_

Round 1
Reviewer 1 Report
In this review El-Jawhari and co-workers address the main variables of bone marrow mesenchymal stem cells (MSCs) mainly in terms of number, proliferation, and differentiation ability that affects their regenerative potential in osteonecrosis. In particular, the authors explain the main function of MSCs to counteract the damage of osteonecrosis and the modifications that occur in MSCs in several physiological, as senescence, and pathological, such as diabetes, lupus, and other disease, conditions which favor osteonecrosis and that may affect their regenerative potential. Overall, the review is well organized and provides interesting information about MSCs biology. A revision of English is recommended since there are some sentences that seems incorrect and typos, as in lines 110-11, 247, 337.
Reviewer 2 Report
This review discusses the quality and quantity of BM-MSCs in relation to the etiological conditions of osteonecrosis such as sickle cell disease, Gaucher disease, alcohol, corticosteroids, Systemic Lupus Erythematosus, diabetes chronic renal disease, and chemotherapy. Although the topic is of interest to the community, the manuscript needs further significant improvements to be considered for publication. The images and schemes are very basic, I would suggest adding more mechanistic and detailed images and schemes to the manuscript. Some parts are missing in the manuscript, for example, what is the role of the extra cellular matrix and the developments by age? See the article "Aging and Osteoarthritis: Central Role of the Extracellular Matrix", how BM-MSCs can contribute in this regard? The authors cited a lot of their own articles in this manuscript, I can see some of them are not perfectly related! Please revise and replace these references with the appropriate ones.
Reviewer 3 Report
This is a well-written and comprehensive review on the use of BM-MSCs for treating osteonecrosis, and how different factors such as age and other disease/lifestyle conditions can affect the MSCs, together with implications for treatment. The review has good structure and includes an impressive collection of relevant studies. Below are some comments to consider that could help further improve the review.
- In Section 2 “MSC therapy for osteonecrosis”, the heading suggests that this section will discuss the current progress in clinical MSC therapy for osteonecrosis. However, this section is mostly about research approaches in development and which have been tested in animal models. While the information that is currently here is relevant, the text should clarify at the start of the section that it is focusing on experimental approaches rather than those routine used in a clinical setting. Furthermore, a sub-section should be included in section 2 to present an overview on the current clinical progress of using BM-MSCs for treating osteonecrosis. If there are relevant systematic or narrative reviews published in this space in the last 5 or so years, these should be cited here and a summary of their findings described.
- In sections 3 to 11, the different factors that could affect MSC quality and quantity in the context of treating osteonecrosis were comprehensively described. However, some of these sections were difficult to follow as the text jumped between preclinical/experimental studies (e.g. in animals, or relating to molecular pathways) and clinical studies involving human subjects/donors. It would make the text much easier to follow if a distinction was made between the preclinical and clinical evidence, perhaps by organising them into sub-sections.
- The conclusions sections should be expanded into a “challenges and perspectives” section or similar. The information already included there is relevant, but it would benefit from a more specific discussion of the challenges faced in satisfactory treatment of osteonecrosis using BM-MSCs in the clinical setting, particularly given the insights provided in this review regarding the different influencing factors/conditions. Perspectives on possible advances in other fields that could help address these challenges in the future should be discussed, e.g. developments in iPSC-derived MSCs, 3D printing/bioprinting to produce more anatomically/functionally relevant scaffolds/tissue, nanomedicine for improved spatial and temporal control of growth factor delivery, methods to control MSC senescence, etc.
Round 2
Reviewer 2 Report
The authors successfully addressed the comments.